# Assembly mechanism of the pleomorphic immature poxvirus scaffold

Jaekyung Hyun [1,2✉], Hideyuki Matsunami [1], Tae Gyun Kim[1,4] & Matthias Wolf [1,3✉]

In Vaccinia virus (VACV), the prototype poxvirus, scaffold protein D13 forms a honeycomb-like lattice on the viral membrane that results in formation of the pleomorphic immature virion (IV). The structure of D13 is similar to those of major capsid proteins that readily form icosahedral capsids in nucleocytoplasmic large DNA viruses (NCLDVs). However, the detailed assembly mechanism of the nonicosahedral poxvirus scaffold has never been understood. Here we show the cryo-EM structures of the D13 trimer and scaffold intermediates produced in vitro. The structures reveal that the displacement of the short N-terminal α-helix is critical for initiation of D13 self-assembly. The continuous curvature of the IV is mediated by electrostatic interactions that induce torsion between trimers. The assembly mechanism explains the semiordered capsid-like arrangement of D13 that is distinct from icosahedral NCLDVs. Our structures explain how a single protein can self-assemble into different capsid morphologies and represent a local exception to the universal Caspar-Klug theory of quasi-equivalence.

[1] Molecular Cryo-Electron Microscopy Unit, Okinawa Institute of Science and Technology Graduate University, 1919-1 Tancha, 904-0495 Onna-son, Okinawa, Japan. [2] Department of Convergence Medicine, School of Medicine, Pusan National University, 50612 Yangsan-si, Gyeongsangnamdo, Republic of Korea. [3] Institute of Biological Chemistry, Academia Sinica, 128 Academia Road Sec. 2, 115 Taipei, Taiwan. [4] Present address: Center for Vaccine Commercialization, R&D Planning Team, Gyeongbuk Institute for Bio Industry, 36618 Andong-si, Gyeongsanbukdo, Republic of Korea.
✉email: jhyu002@pusan.ac.kr; matthias.wolf@oist.jp

Poxviruses are large, enveloped, double-stranded DNA viruses that infect humans, other vertebrates, and arthropods. Viral infection typically causes skin lesions but can also prove fatal, as in the case of smallpox. Despite eradication of smallpox, zoonotic outbreaks of smallpox-like diseases urge thorough understanding of the poxvirus replication cycle to facilitate the development of strategies for prevention and treatment[1].

Poxviruses share a common phylogeny and replication pathways similar to those of NCLDVs[2]. A common feature of NCLDVs is the presence of major capsid proteins with a double-jelly-roll motif that assemble into icosahedral capsids. Notable exceptions are poxviruses, ascoviruses, and pandoraviruses, which exhibit variable capsid diameters and do not follow canonical symmetry[3,4]. In Vaccinia virus (VACV), copies of the double-jelly-roll protein D13 bind to the viral membrane via interaction with its partner protein A17 and assemble into a scaffold surrounding the viral membrane[5,6]. This process leads to the formation of a spherical immature virion (IV). Subsequent proteolytic processing of A17 releases the D13 scaffold from the viral membrane, leading to a dramatic transformation into the brick-shaped mature virion (MV). Unbound to the viral membrane, D13 exist as trimers and is found forming aggregated hexagonal patches and rodlets in the viral factory. Only with the expression of viral late proteins and in association with the viral membrane, D13 trimers assemble into a crescent-shaped honeycomb lattice followed by complete formation of the spherical immature virion upon DNA uptake[7,8]. However, in vitro assembly of D13 demonstrated formation of spherical particles with the morphology comparable to authentic IV scaffolds[9]. This suggests that the intrinsic curvature induced by D13 self-assembly is playing a major role in governing the shape and size of IVs[7,9,10]. Hence, detailed molecular insights into the intermolecular D13 interactions are needed to understand the scaffold assembly mechanism.

The 62-kDa D13 protein exists as trimers both in vivo and in vitro, and structures of the recombinantly expressed D13 trimer have been determined by X-ray crystallography[8,9,11,12]. D13 adopts a double-jelly-roll structure composed of 8 antiparallel β-strands. A head domain is inserted between β-strands of the C-terminal jelly roll. Recently, the structure of D13 in complex with the antibiotic rifampicin was determined by X-ray crystallography, illustrating the mechanism by which the antiviral drug prevents binding of A17, occupying the phenylalanine-rich pocket on the 3-fold axis of the D13 trimer[12]. Inhibition of the D13-A17 interaction would impair recruitment of D13 onto the viral membrane, leading to aberrant virion formation, as previously demonstrated by mutagenesis[13–17]. However, none of the crystallographic packing arrangements from any of the reported structures represent the honeycomb-shaped lattice observed on the surface of authentic IV, and intertrimer interactions that govern D13 assembly have been poorly resolved. Low-resolution electron microscopy and three-dimensional (3D) reconstructions obtained from negatively stained two-dimensional (2D) planar arrays of D13 and its orthologue, orfv075 protein, from orf virus provided the overall organization of trimers in the context of the honeycomb-shaped lattice[9,18]. Nevertheless, the molecular details of the intertrimer interface were inconclusive in these studies. Here we show the cryo-EM structures of D13 trimers, trimer doublets, and the tubular assembly of an expanded honeycomb lattice produced in vitro. The structures reveal that the short N-terminal α-helix is critical for the initiation of D13 self-assembly. The continuous curvature of the immature virion is mediated by electrostatic interactions that induce torsion between the trimers. The results explain the mechanism of semiordered capsid-like arrangement of D13, which is distinct from

homologous capsid proteins of icosahedral NCLDVs. Our structures demonstrate how a single protein can self-assemble into different capsid morphologies induced by a small conformational change of a peripheral helix.

## Results

**Structure of D13 trimer.** We purified recombinant D13 trimer protein for structure determination using single-particle cryo-electron microscopy (cryo-EM). Initial attempts were hampered by problems associated with preferred particle orientation and spontaneous protein assembly into honeycomb-like crystalline patches that consist of rings of five or six trimers (Supplementary Fig. 1). This problem was mitigated using holey EM grids coated with a graphene oxide support film (Supplementary Fig. 2a), and the cryo-EM structure of the D13 trimer was determined at 2.3 Å resolution (Fig. 1a and Supplementary Fig. 2). In our atomic model, we followed the convention for the jelly-roll fold with BIDG-CHEF β-strand-order (Fig. 1b). The refined atomic model based on this structure was largely identical to previously reported crystal structures (average RMSD 0.495 Å)[9,11,12]. However, the presence of the short N-terminal tail α-helix (aa 1-10) was inconsistent with previously reported models. In the first two crystal structures, electron density for this helix was only observed in the D13_{D513G} mutant but not in wild type protein[9,11], whereas subsequently reported crystal structures contain the

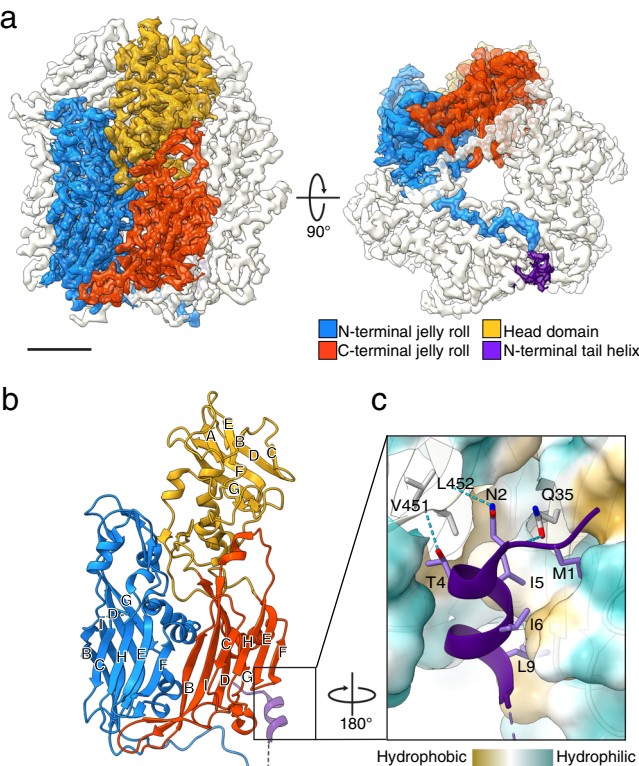

a

b

c

Hydrophobic ▨ Hydrophilic

- N-terminal jelly roll
- C-terminal jelly roll
- Head domain
- N-terminal tail helix

**Fig. 1 Cryo-EM structures of the VACV D13 trimer. a** Isoelectron potential surface of the cryo-EM reconstruction of the D13 trimer contoured at 3.0σ above average. The map resolution is 2.3 Å. The fitted atomic model of one monomer is shown in stick representation within the transparent isosurface. **b** Schematic architecture of the D13 monomer depicted in ribbon representation, composed of two jellyroll domains and one head domain. β-strands in each domain are labeled in alphabetical order, based on ascending amino acid sequence. **c** A close-up view of the N-terminal tail α-helix model stabilized in the hydrophobic pocket of a neighboring monomer and hydrogen bonds (dashed cyan line). The Conolly surface of that monomer is colored according to its molecular lipophilicity potential.

corresponding density in only two of the three subunits[12]. Our cryo-EM map showed intact N-terminal tail α-helices in all three subunits, with or without imposition of 3-fold symmetry, although reduced local map resolution and the increased B-factor suggested that the helix was prone to dislocation (Supplementary Fig. 2f–h). In our structure, the hydrophobic face of the N-terminal helix (M1, I5, I6 and L9) was held in a pocket of surrounding hydrophobic residues from the adjacent subunits. Additional hydrogen bonds were identified between the side chains of T4, N2, Q35 and the carboxylic groups of V451, L452, and N2 (Fig. 1c). The contact between the N-terminal helix and the interaction pocket is nearly identical when compared to previously determined crystal structures (PDB 3SAM, 2YGC, 6BEI)[9,11,12], with marginal discrepancy of the contact surface area (Supplementary Fig. 3a).

**Intertrimer interactions in D13 trimer doublet**. Image processing of D13 trimers revealed a small subpopulation of adjacent trimers bound together, forming doublets of trimers. Further 2D and 3D classifications enabled segregation of doublet particle images from the trimer singlets. The scarcity of intermediate angular views within the dataset was supplemented by additional micrographs acquired from tilted specimen grids without the graphene oxide support film (Supplementary Fig. 4a). Although the resolution of the resulting 3D reconstruction (3.9 Å on average) was not fully isotropic (Supplementary Fig. 4e), it allowed unambiguous positioning of an atomic model for identification of the intertrimer interfaces. There are two major interfaces that mediate doublet formation (Fig. 2a). The first is located between one of the head domains on each of the two trimers. Although map resolution at this interface was insufficient

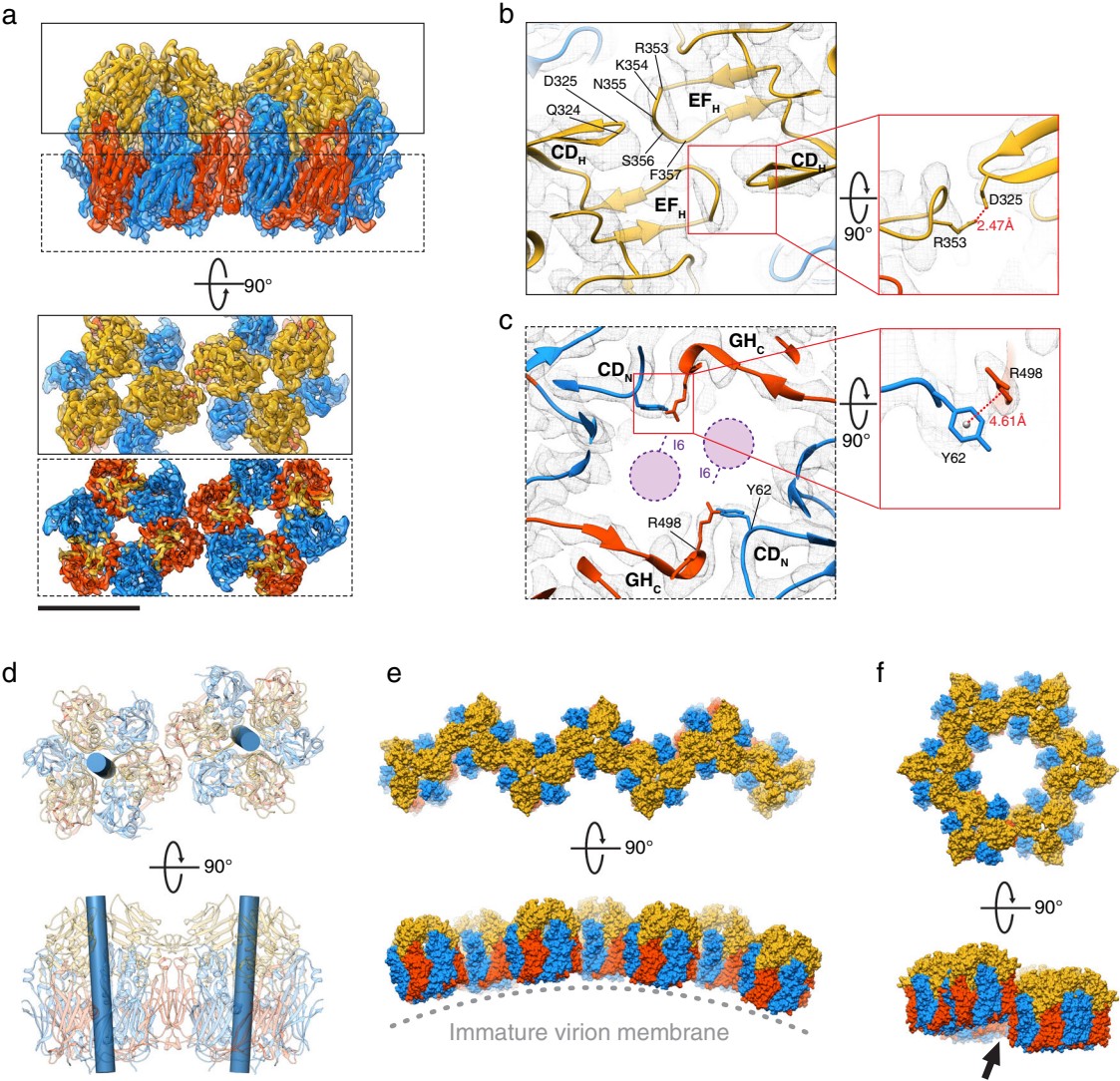

**Fig. 2 Cryo-EM structures of the VACV D13 trimer doublet. a** Isoelectron potential surface (contoured at 3.0σ above average) of D13 trimer doublet with the docked molecular model. Intertrimer contacts are revealed in slabs (boxes with bold and dashed lines) extracted at different clipping planes. **b** Head-to-head contact between the trimers in the doublet model superposed on the cryo-EM map. A potential salt bridge interaction between R353 and D325 is shown in a magnified view in a red box. **c** Base-to-base contact between trimers in the doublet model superposed on the cryo-EM map. Positions of the missing N-terminal tail α-helix in the doublet structure are depicted as purple circles, with its isoleucine (I6) rotamer indicated by a protruding dashed line. The cation-π interaction between Y62 and R498 is shown in the red box. **d** Relative position of neighboring trimers in the doublet as shown by cylinders, indicating the 3-fold symmetry axis of each trimer. The torsion angle between the axes is 8.0°. **e, f** Linear and planar expansions of the D13 doublet interface. The side view of the linear expansion exhibits convex curvature with the trimer base facing the center of the arch. Planar expansion results in a ring of six trimers, but the ring does not close (arrow) due to geometrical clash. Scale bars, 5 nm.

to clearly define side chain interactions, it was possible to trace the loops extending toward the neighboring trimer (Fig. 2b). These include an extended loop that connects β-strands E and F (EF_H loop) and a hairpin loop between β-strands C and D (CD_H loop) in the head domain. The interface is clustered with polar (N355, Q324, S356) and charged amino acid residues (D325, R353, K354) around the two-fold symmetry axis. Side chains of R353 in the EF_H loop and D325 in the CD_H loop extend toward each other, suggesting salt bridge interactions. Another interface is located at the base of the trimers, which would be proximal to the viral membrane in the IV. At this interface, the major difference of doublet structure in comparison to the singlet trimer is the unambiguous absence of the N-terminal tail α-helix (Fig. 2c and Supplementary Fig. 3b). The interface involves the loop that connects β-strands C and D of the N-terminal jelly-roll (CD_N loop) and the loop connecting β-strands G and H of the C-terminal jelly-roll (GH_C loop) in the neighboring trimer. Unlike the head-to-head interface, this interaction occurs pairwise between two monomers in each of the two trimers. The clearly resolved features in the reconstructed electron potential map between Y62 and R498 suggests cation-π interactions as the main stabilizer at this interface. We found Y62 in close proximity to I6 of the N-terminal tail α-helices in the singlet trimer structure, suggesting that the hydrophobic interaction between these residues may have been altered in favor of intertrimeric interaction in the doublet structure, upon dislocation of the N-terminal tail α-helix. Furthermore, the doublet is characterized by a slight twist of the trimer axes defined by a torsion angle of 8.0°, leading to an arched arrangement when the doublets are linearly expanded (Fig. 2d, e). The bases of D13 trimers face the center of the arch, which supports the idea that remodeling of the viral membrane into a convex curvature is induced by the self-assembled D13 scaffold. However, planar expansion of the doublet prohibits closure of the dome-shaped curvature, suggesting that additional intertrimer interfaces or oligomers are involved in the formation of spherical poxvirus scaffolds (Fig. 2f).

**Critical role of the N-terminal α-helix in poxvirus scaffold assembly.** The absence of the N-terminal tail α-helix in the doublet structure suggests that its displacement likely helps to initiate intertrimer interactions, promoting ordered assembly of oligomers and leading to formation of spherical IV-like particles. To test this hypothesis, we used in vitro D13 self-assembly. The protein solution turned turbid after dialysis into low-salt buffer, but examination by transmission electron microscopy (TEM) showed aggregated honeycomb-like patches and rodlets without any IV-like particles (Fig. 3a). Under the same conditions, the D13 protein with an N-terminal His_6-tag and an extended linker (28 amino acid residues) assembled into spherical particles with an average diameter of approximately 340 nm, revealing a distinctive honeycomb-shaped pattern (Fig. 3b, Supplementary Fig. 5 and Supplementary Movie 1). These particles clearly resemble the size and shape of the authentic VACV IV scaffold described by deep-etch electron microscopy[19]. Although these observations were previously reported, the rationale behind the assembly discrepancy between tagged and untagged D13 could not be explained[9]. To investigate whether the N-terminal tail α-helix is indeed responsible for this observation, we determined the structure of the His_6-tagged D13 protein at 2.6 Å resolution by cryo-EM. As expected, the N-terminal tail helix feature was absent in the reconstructed map. This is most likely due to the hydrophobic residues linking D13 (such as Y-4 and F-3) and the His_6-tag, which reach into the hydrophobic pocket as seen in a crystal structure (PDB ID 3SAM)[9] and thereby perturb the position of the N-terminal helix (Supplementary Fig. 3c).

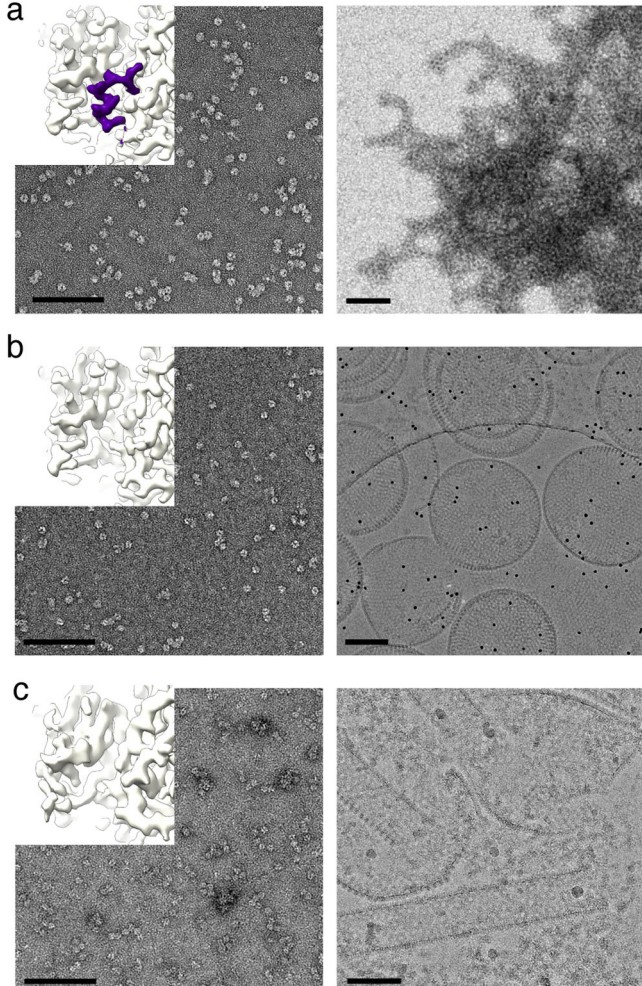

**Fig. 3 Assembly of D13 requires displacement of the N-terminal tail α-helix.** Transmission electron micrographs of **a** wtD13, **b** His_6-tagged D13 and (**c**) D13_{18-548}, before (left column) and after (right column) low-salt buffer-mediated assembly. Images are of negatively stained samples, except in the right column of **b** and **c** where images of vitrified samples captured by cryo-EM are shown for clear morphological observation. Insets show details of the respective contoured cryo-EM reconstructions. 130,384, 173,354 and 156,813 particle images were used to reconstruct the maps shown in **a**, **b**, and **c**, respectively. Their structures reveal that the wild-type trimer has an intact N-terminal tail helix (colored purple), whereas the corresponding density is missing in His_6-tagged protein and in the truncation mutant. Proteins exist as homogenous trimers in 600-mM NaCl storage buffer. Reduction of salt concentration to 150 mM leads to aggregation in case of wtD13, to spherical assemblies for His_6-tagged D13, or to a mixture of tubular and spherical assemblies for D13_{18-548}. The spherical particles in **b** were sometimes imaged as concentric shells containing several layers, or incompletely closed structures. Isoelectron potential surfaces were contoured at 3.0σ above average. Map resolutions are 2.3 Å, 2.6 Å and 4.1 Å (**a**, **b**, **c**, respectively). Scale bars, 100 nm.

Furthermore, we expressed and purified mutant protein (D13_{18-548}) with a truncation of 17 N-terminal amino acid residues, including the N-terminal tail α-helix. Our cryo-EM reconstruction of the mutant displayed no significant change from wild-type D13 (wtD13) or D13 with an intact N-terminal His_6-tag. This mutant assembled into tubular and spherical structures that exhibit a honeycomb-like lattice (Fig. 3c) under low-salt conditions in a fashion similar to the His_6-tagged protein, thereby

confirming the essential function of the N-terminal tail helix for D13 self-assembly. Based on the observation that both spherical and tubular assembly products co-exist under the same assembly condition, these objects are believed to be variants of curved assembly.

**Intertrimer interactions in tubular D13 assembly.** To investigate intertrimeric interactions in the context of higher-order

assembly, tubular assembly products of the N-terminal truncation mutant $D13_{18-548}$ were subjected to cryo-EM and helical reconstruction (Supplementary Fig. 6a–d). Refined helical parameters indicated a 3-start helix with a 33.9-Å rise and a 77.0° twist. A 3D reconstruction at 7.3 Å resolution exhibited a continuous honeycomb-shaped lattice rolled into a tube (Fig. 4a). Single-particle analysis of symmetry-expanded, signalsubtracted particle images of trimer sextets resulted in improved cryo-EM map resolution at 3.9 Å, enabling further analysis of intertrimer

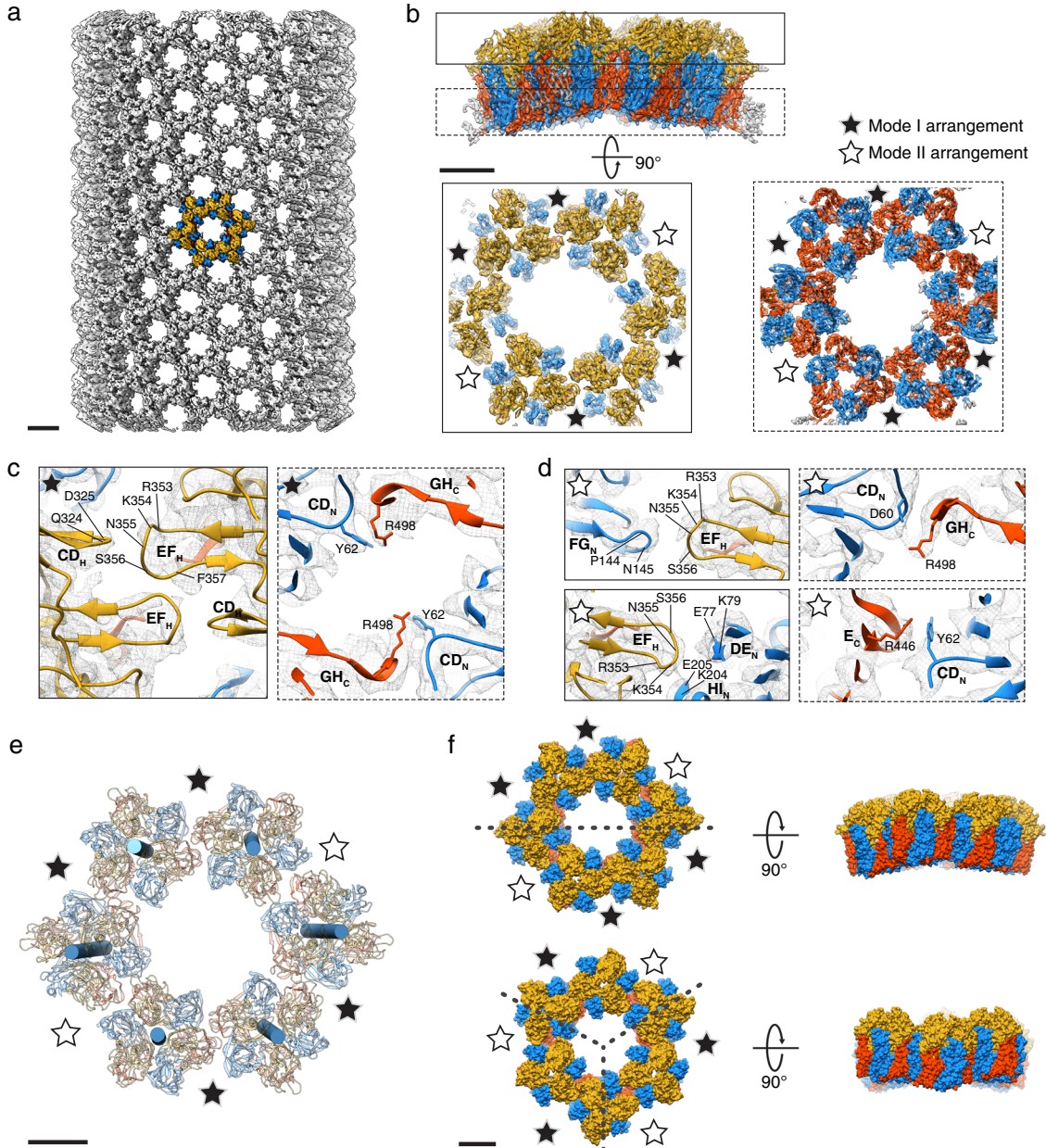

**Fig. 4 Structure of a tubular assembly of D13 trimers. a** 3D reconstruction of the tubular D13 assembly at 7.3 Å resolution. The map was contoured at 5.0σ above average. A hexameric ring of trimers is colored based on the scheme in Fig. 1. **b** 3D reconstruction of a signalsubtracted D13 trimer sextet at 3.9 Å resolution (isosurface contoured at 5.0σ) with a superposed atomic model. Intertrimer contacts are revealed within slabs through the reconstruction extracted at different clipping planes (boxes with bold and dashed lines). **c, d** Cartoon representation of a molecular model depicting loops and amino acid residues involved in intertrimer interfaces. The mode I arrangement features two-fold symmetry, whereas contact points in the mode II arrangement are asymmetric. **e** Relative position of neighboring trimers in the sextet, as shown by cylinders indicating the 3-fold symmetry axis of each respective trimer. **f** Curvature of the D13 trimer sextet found in tubular assembly (upper panel) and manually modeled ring of six trimers with alternating mode I and II arrangements (lower panel). The curvature is unidirectional in tubular assembly along the direction depicted by the dashed line whereas alternating modes of arrangements allow for full closure of the ring while generating the curvature around the 3-fold symmetry axis. Scale bars, 10 nm in **a** and 5 nm in **b**, **e**, **f**, respectively.

interfaces (Fig. 4b and Supplementary Fig. 6e–h). The structure exhibits two alternating modes of intertrimer arrangements, which are mostly stabilized by electrostatic interactions (Supplementary Fig. 7) and agree with the self-assembly behavior of D13 with varying salt concentrations of the buffer solution. The first mode (mode I) is similar to the doublet structure in which the interaction between the head domains appears most extensive between the $EF_H$ and $CD_H$ loops. The base-to-base interface involves cation-π interactions between Y62 of the $CD_N$ loop and R498 of the $GH_C$ loop (Fig. 4c). The second mode (mode II) also implicates two intertrimer interfaces. However, in this arrangement, two monomers in each trimer are involved, forming pairwise interactions (Fig. 4d). In one moiety, contacts between the $EF_H$ and $FG_N$ loops (P144 and N145) appeared most prominent. In the other moiety, the $EF_H$ loop is in close proximity to charged residues in a short β-strand in the $DE_N$ loop (E77 and K79) and a short α-helix in the $HI_N$ loop (E205 and K204). Symmetric Y62-R498 interactions were no longer found in the base-to-base interface of the mode II arrangement. One moiety appears to involve a cation-π interaction between Y62 and R446 in β-strand E of the C-terminal jelly roll ($E_c$), replacing R498 in the mode I arrangement. The other moiety exhibits close contact between R498 of one monomer and Y62 and D60 of the other monomer. Residues R353 and Y62 were critical in all modes of arrangements for head-to-head and base-to-base interfaces, respectively. This was confirmed by mutation of these residues to alanine, resulting in assembly incompetence in low-salt buffer (Supplementary Fig. 8). Each trimer was positioned with a slight torsion in relation to neighboring trimers in the sextet. Torsion angles ranged from 7.6° to 9.4° and from 12.5° to 13.5° in mode I and mode II arrangements, respectively (Fig. 4e). The torsion angle varied among mode I arrangements, indicating slight plasticity. While the twist direction was consistent for mode I arrangements, it alternated depending on the asymmetric contact points between the trimers. Variability of the intertrimer torsion angle and direction found in the helical assembly suggests a possible continuous curvature in all directions, inducing the spherical morphology observed in authentic VACV IV. To examine this, two types of intertrimer arrangements were alternatively modeled instead of two consecutive mode I arrangements, followed by a mode II arrangement, as originally found in the tubular assembly. This allowed for closure of a crown-shaped ring of six trimers (Fig. 4f). However, expansion of the sextet failed to generate a continuous spherical morphology.

## Discussion

Poxvirus scaffolds consist of loosely connected rings of six trimers in contrast to tightly packed capsomers of icosahedral NCLDV capsids that require additional minor capsid proteins and accessory proteins[20–23]. Poxviruses require concerted interactions between viral late proteins for the formation of the capsid-like scaffold in vivo[6], with the exception of the D13$_{D513G}$ mutation that results in stacks of planar honeycomb lattices in the absence of viral late proteins[8]. However, we show that scaffold morphology is mainly governed by self-assembly of D13 without accessory proteins. The charge distribution on the surface of the D13 trimer roughly resembles that of the capsomers of NCLDVs and hence is likely to follow their capsid assembly scheme based on electrostatic interactions[24]. The intertrimeric torsion and its variation are unique features of the poxvirus scaffolding protein. We revealed that the curvature that results from alternating torsion between the D13 trimers forms the basis of semiregular, spherical immature poxvirus particles. Mild curvature between the capsomers has been described for African swine fever virus (ASFV) major capsid protein, although such curvature depends

on the microenvironment within the icosahedron and is dissimilar to continuous curvature in the case of poxvirus scaffolds[21].

We found that the N-terminal short tail α-helix of D13 is critical for in vitro assembly into ordered structures. We propose that repositioning of the N-terminal tail α-helix away from D13 toward the surface of the lipid bilayer may act as a molecular trigger that initiates scaffold assembly in vivo. This scenario implies that D13 assembly into a scaffold occurs in situ only when bound to the viral membrane via the D13-A17 interaction, but not before, thereby preventing the virion assembly sequence without a lipid membrane. This is supported by cellular cryo-electron tomography (cryo-ET) of immature poxvirus, which showed that intact scaffold formation requires association with a lipid membrane[7]. N-terminal helices have been found to act as anchors on proximal lipid membranes in double-jelly-roll capsid proteins of dsDNA bacteriophages and African swine fever virus (ASFV)[25–27]. However, such interactions were only proposed to stabilize the capsid and not to initiate assembly.

The N-terminal helix and residues involved in the intertrimeric interactions of D13 are widely conserved among chordopoxviruses, suggesting a common scaffold mechanism. In contrary, this conservation is poor in entomopoxviruses; most notably the absence of the N-terminal helix (Supplementary Fig. 9). The intertrimer interfaces are distal from the rifampicin binding site and from residues that confer rifampicin resistance in VACV, indicating that these intertrimeric interactions are independent of drug binding[12,14] (Supplementary Fig. 10a). Aspartic acid residues 513 (D513) from neighboring trimers were within 7 Å along the 2-fold symmetry axis in the mode I arrangement, but mutation D13$_{D513G}$, which leads to abrogation of the curved lattice[8], could not be explained by our structure (Supplementary Fig. 10b).

Nearly 60 years ago, Donald Caspar and Aaron Klug introduced their theory of quasi-equivalence of macromolecular assemblies, which predicts arrangements of protein subunits at the surface of icosahedral virus particles in a regular lattice, parametrized by a triangulation number (T). The Caspar-Klug theory has been instrumental in understanding the molecular symmetry of icosahedral viruses and many other macromolecular assemblies. With the advent of cryo-EM at near-atomic resolution, a number of deviations from ideal symmetry have been discovered[28–30]. The VACV scaffold is an example of such a deviation, in which D13 is uniquely responsible for generating various building blocks and continuous curvature. The observation of pentameric tiling (Fig. 5b and Supplementary Fig. 1) supports the hypothesis that the underlying symmetry of the VACV scaffold as a result of D13 assembly is an icosahedral lattice, but local deviations result in an imperfect, yet semi-regular scaffold (Fig. 3b, Supplementary Fig. 5 and Supplementary Movie 1). The N-terminal helix represents a mobile element that regulates the initiation of scaffold assembly and electrostatic intertrimer interactions, resulting in different local symmetries that follow the principle of quasi-equivalence within a patch of common assembly mode, but not throughout the entire scaffold (Fig. 5c). Using only a single gene product, such locally irregular transient scaffold assembly may serve as an economic and efficient way of packaging the large poxvirus genome in its immature virion. The interactions identified in our study may inspire the design of a new generation of assembly inhibitor drugs (e.g., peptidomimetics based on the N-terminal helix) and further structural investigation of other pleomorphic viral capsids, such as HIV.

## Methods

**Protein expression and purification**. The gene encoding D13 of Western Reserve strain Vaccinia virus in the pPROEX-Hta vector was expressed in BL21(DE3) cells

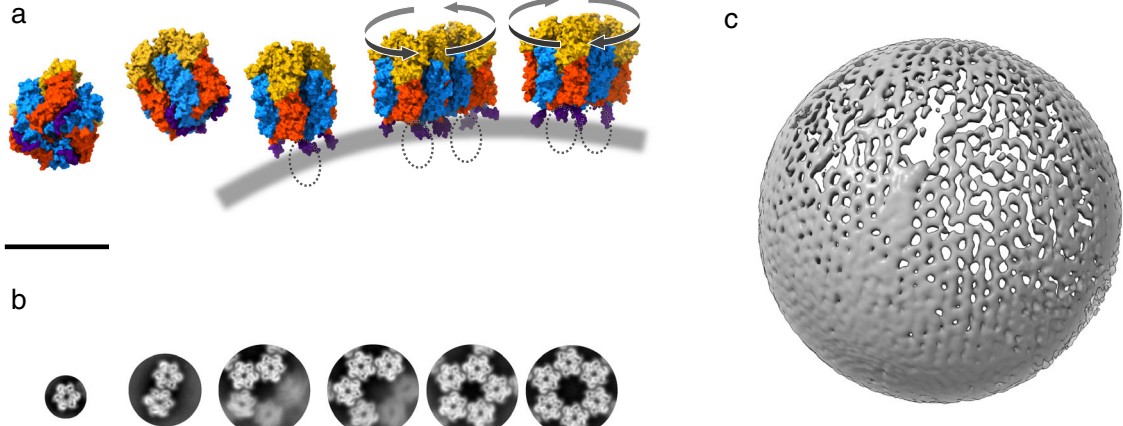

**Fig. 5 Capsid-like assembly hypothesis of the immature VACV scaffold in vivo. a** Schematic illustration of D13 assembly initiation. D13 exists as individual trimers in the cytoplasm. Once D13 is recruited onto the surface of a viral membrane (depicted by a curved gray line) through an interaction with A17 (depicted as dashed ellipses), the N-terminal helix (purple) is dislocated away from D13 and exposes the base-to-base intertrimer interface. D13 trimers can subsequently dimerize with two distinct intertrimer torsions (clockwise and counterclockwise arrows), inducing curvature. **b** Gallery of D13 oligomers that are basic building blocks of the honeycomb-shaped lattice. Class averages were generated from electron micrographs of honeycomb-like patches (Supplementary Fig. 1). **c** Expansion of the honeycomb-like lattice into a spherical scaffold. The cryo-electron tomography reconstruction shows a spherical IV scaffold-like particle (diameter 340 nm) with a typical honeycomb-like lattice. Although the limited resolution precluded detailed modeling of the scaffold, it is likely that additional patterns of oligomers such as pentameric building blocks or different combinations of intertrimer arrangements are required to form the authentic immature VACV scaffold. Scale bars, 10 nm in **a**, **b**, and 100 nm in **c**, respectively.

and purified by immobilized metal affinity chromatography (IMAC) using His-TrapHP (Cytiva). The N-terminal His$_6$-tag was removed by TEV protease digestion, and the products were further purified by IMAC to remove the remaining tagged protein. TEV protease digestion was omitted for the production of His$_6$-tagged protein used in D13 self-assembly experiments. The protein was further purified by size exclusion chromatography in a buffer containing 50 mM Tris-HCl (pH 8.0), 600 mM NaCl, 50 mM L-arginine, 50 mM L-glutamic acid and 2 mM β-mercaptoethanol.

**Mutagenesis.** The N-terminal truncation mutant (D13$_{18-548}$) was generated by PCR using a forward primer lacking the leading 51 bases of the D13 DNA sequence. Point mutations R353A and Y62A were introduced to wtD13 and D13$_{18-548}$ using a Q5 site-directed mutagenesis kit (New England Biolabs Inc.). Expression and purification procedures for the mutant were identical to those of wtD13. Primers used for mutagenesis are listed in Supplementary Table 1.

**D13 self-assembly.** A 100-μL aliquot of purified protein at approximately 2 mg/mL was dialyzed in buffer containing 10 mM Tris-HCl (pH 8.0), 150 mM NaCl, and 2 mM β-mercaptoethanol at 4 °C for at least 12 h. The resulting turbid solution was centrifuged at 18,363 × g. Then, 80 μL supernatant was removed, and the pellet was resuspended using the remaining solution. The pellet sample was negatively stained with 2% uranyl acetate and examined using TEM to confirm the assembly. The remaining supernatant was checked by TEM to confirm the presence of homogeneous trimers. This preparation was used for cryo-EM of singlet trimer particles.

**Negative staining and transmission electron microscopy.** Routine TEM examinations were performed using negatively stained samples. A total of 5 μL of protein at approximately 0.01 mg/mL were loaded onto freshly glow-discharged EM grids with continuous carbon support films. A total of 90 s were allowed for sample adsorption, and then grids were washed with three droplets of distilled water. A total of 5 μL of 2% uranyl acetate solution were loaded onto grids, followed by 60-s incubation. A piece of filter paper was used to blot excess stain solution and grids were air-dried. Specimen grids were examined using a Talos L120C TEM equipped with a Ceta CMOS detector (Thermo Fisher Scientific (TFS)), operating at 120 kV acceleration voltage.

**Cryo-electron microscopy.** For wtD13 trimers, His$_6$-tagged D13 trimers and D13$_{18-548}$ trimers, 3 μL of protein at 0.12 mg/ml were loaded onto holey EM grids (Quantifoil R1.2/1.3 Cu300, Quantifoil Micro Tools GmbH) treated with graphene oxide film flakes (Sigma Aldrich). The sample was vitrified on a Vitrobot Mark IV (TFS) operating at 4 °C and >90% relative humidity. wtD13 image data were acquired with a Titan Krios TEM (TFS) operating at 300 kV at a nominal magnification of 155,000×, corresponding to 0.518 Å/pixel at the specimen level, with the defocus ranging between 0.5 μm and 1.5 μm. Spot size 8, C2 aperture 70 μm, OL aperture 100 μm and 1.8 μm illumination area were used. 50 movie fractions

collected on a Falcon 3EC direct electron detector in electron counting mode using EPU software (TFS), with a total electron dose of 50 e⁻/Å² and dose rate of 1.6 e⁻/Å²/s. His$_6$-tagged D13 image data were collected with a Titan Krios TEM (TFS) operating at 300 kV at a nominal magnification of 155,000×, corresponding to 0.518 Å/pixel at the specimen level, with the defocus ranging between 0.5 μm and 1.5 μm. Spot size 8, C2 aperture 70 μm, OL aperture 100 μm and 1.8 μm illumination area were used. 50 movie fractions were collected on a Falcon 3EC direct electron detector in electron-counting mode using EPU software (TFS), with a total electron dose of 50 e⁻/Å², and dose rate of 1.6 e⁻/Å²/s. D13$_{18-548}$ data were collected on a Talos Arctica TEM (TFS) operating at 200 kV at a nominal magnification of 92,000×, which corresponds to 1.12 Å/pixel at the specimen level, with defocus ranging between 0.6 μm and 1.2 μm. The spot size was 9. A 70-μm C2 aperture and 100-μm OL aperture were used. Exposures were acquired as movies of 50 dose fractions on a Falcon 3 direct electron detector in electron counting mode using EPU software (TFS), with a total electron dose of 50 e⁻/Å² at a dose rate of 0.6 e⁻/Å²/s.

For D13 doublets, multiple datasets were collected to enrich the angular orientation of particle images. The first two datasets were collected from a grid prepared for the D13 trimer singlet using holey carbon grids with additional graphene oxide film. For the third and fourth datasets, samples were prepared on gold grids (UltrAuFoil R1.2/1.3 300, Quantifoil Micro Tools GmbH) without a graphene oxide support film. Vitrification methods were the same for all grid preparations. For the third and fourth datasets, in which D13 trimers were preferentially oriented and partially assembled, the microscope stage was tilted to 30° and 45°, respectively. Image data were collected on a Talos Arctica TEM (TFS) operating at 200 kV at a nominal magnification of 92,000×, which corresponds to 1.12 Å/pixel at the specimen level, with defocus ranging between 0.5 μm and 5.0 μm. The spot size was 9 or 10. A 50- or 70-μm C2 aperture and 100-μm OL aperture were used. Exposures were acquired as movies of 50 dose fractions on a Falcon 3 direct electron detector in electron counting mode using EPU software (TFS), with a total electron dose of 50 e⁻/Å² at a dose rate between 0.6 and 0.9 e⁻/Å²/s.

For tubular D13 assembly, 3 μL of resuspended pellet from the assembly solution were loaded onto a holey EM grid (Quantifoil R2/2 Cu300, Quantifoil Micro Tools GmbH). Vitrification methods were the same as above. Data were acquired on a Titan Krios TEM (TFS) operating at 300 kV in EFTEM mode (Gatan Quantum 968) at a nominal magnification of 105,000× corresponding to 1.39 Å/pixel at the specimen level, with defocus ranging between 0.5 and 2.5 μm. Spot size 7, a 70-μm C2 aperture and 100-μm OL aperture were used. 50 dose fractions/movie were recorded with a K2 Summit direct electron detector (Gatan) and EPU software (TFS), with a total electron dose of 50 e⁻/Å² at a dose rate of 5 e⁻/Å²/sec.

For cryo-ET of spherical IV-like particles, assembly solutions that resulted from introducing His$_6$-tagged D13 into low-salt buffer were mixed with 10-nm gold fiducial marker (AURION) in a 4:1 (v:v) ratio. 3 μL of the mixture were loaded onto a nonglow-discharged holey EM grid (Quantifoil R2/2 Cu300, Quantifoil Micro Tools GmbH) and vitrified on a Vitrobot Mark IV (TFS) at 4 °C and >90% relative humidity. Tomographic data were acquired on a Titan Krios TEM (TFS) operating at 300 kV at a nominal magnification of 45,000× corresponding to 2.26 Å/pixel at the specimen level, defocused between 5.0 and 6.0 μm. The spot size was 9, and a 70-μm CL aperture with a 100-μm OL aperture

and a 1.8-μm beam diameter area were used. A dose-symmetric tomography acquisition scheme was applied on a Falcon 3 direct electron detector in EC mode using Tomography 4 software (TFS) over a ±63° tilt with 3° intervals. The total dose applied to each tomogram was approximately 120 e⁻/Å², and 5 movie fractions were collected for each tilt image at a dose rate of 0.14 e⁻/Å²/s.

**Cryo-EM Image processing**. All single-particle datasets were processed with Relion 3.1 software unless stated otherwise[31–34]. Motion correction and contrast transfer function (CTF) estimation were performed using MotionCor2[35] and CTFFIND4[36], respectively. Statistics of individual datasets can be found in Supplementary Table 2. Cryo-ET data were processed using IMOD. Visual examination of maps and figure preparations were performed with UCSF Chimera[37], Chimera X[38] and PyMol (The PyMol Molecular Graphic System, Version 2.4.0, Schrödinger, LLC).

For wtD13 trimers, 2,168 dose-weighed micrographs were manually inspected. Images without graphene oxide support, poor estimated maximum resolution or strong astigmatism from CTFFIND calculations were discarded. From 1,862 assorted micrographs, particles were automatically picked using 2D class averages of manually picked particles as a template. A total of 668,437 particles were binned by 4 and extracted into 128 pixel squared boxes. The particles were then subjected to two rounds of 2D classification to eliminate poorly aligned particle images (Supplementary Fig. 2a). The resulting 204,563 particles were re-extracted from micrographs with a binning factor of 2 into 256 pixel square boxes. A consensus reconstruction was generated by refinement without imposition of symmetry. Then, 3D classification was performed without particle alignment, from which 130,384 particles that belong in the class exhibiting the best structural details were selected for the final round of refinement (Supplementary Fig. 2b). Further refinements on the assorted particles were performed with C3 symmetry imposition. CTF refinement for magnification anisotropy, optical aberrations and per-particle defocus were performed, followed by Bayesian particle polishing. Final 3D refinement and map sharpening resulted in a reconstruction at 2.25 Å spatial resolution based on 0.143 Fourier shell correlation (FSC) criterion of independently refined halfset reconstructions (Supplementary Fig. 2c–e). The particles that had been selected from the 3D classification were also subjected to 3D refinement without symmetry imposition, CTF refinement and Bayesian polishing. The final resolution of the unsymmetrized 3D reconstruction was estimated at 2.63 Å resolution (FSC = 0.143).

For His₆-tagged D13 trimers, 1,328 dose-weighed micrographs were selected from 1,418 total micrographs after removing micrographs with poor estimated maximum resolution or strong astigmatism. A total of 284,340 particles were semiautomatically picked, binned by 4, and boxed in 128-pixel squared boxes. Two rounds of 2D classifications were performed to remove poorly aligned images. The resulting 216,590 particles were re-extracted from micrographs with a binning factor of 2 into 256-pixel squared boxes, resulting in a sampling interval of 1.036 Å/pixel. 3D classification was performed without symmetry imposition, and only particles that belong to classes with detailed structural features were selected. The resulting 173,354 particles were subjected to 3 rounds of 3D refinement with C3 symmetry. CTF refinement and particle polishing were performed between refinements. The final resolution of the 3D reconstruction was estimated at 2.63 Å (FSC = 0.143).

For D13₁₈₋₅₄₈ trimers, 1,112 dose-weighed micrographs out of 1,227 total micrographs were selected after removing micrographs with poor estimated maximum resolution or strong astigmatism. A total of 743,834 particles were semiautomatically picked, binned by 2, and boxed in 128-pixel squared boxes. A 2D classification was performed to remove poorly aligned images, and 475,652 good particles were re-extracted from unbinned micrographs (1.12 Å/pixel), into 256-pixel square boxes. 3D classification was performed without symmetry imposition and 156,813 particles that belong to classes with detailed structural features were selected. 3 rounds of 3D refinement with C3 symmetry were performed, with CTF refinement and particle polishing between. The final resolution of the 3D reconstruction was estimated at 4.10 Å resolution (FSC = 0.143).

For D13 trimer doublets, four datasets were used (550, 280, 676, and 189 movies each), in which each movie dataset was independently processed for beam-induced motion correction, CTF estimation, and particle autopicking. Particle autopicking was performed using a template 3D map produced by artificially joining two copies of D13 trimer maps. To minimize the effect of reference bias on the high-resolution signal imposed by particle picking, the template was low-pass filtered to 20 Å. A total of 835,797 particles from a combined 1,695 micrographs were binned by a factor of 2 and extracted into 128 pixel squared boxes. Particle images were subjected to 2D class averaging, from which images that belonged to classes with poor structural features were eliminated (Supplementary Fig. 4a). The resulting 746,660 particles were subjected to 3D classification without symmetry imposition using an initial 3D reference generated from image data. The majority of particles partitioned into 3D classes with a pronounced trimer singlet featuring an adjacent ghost-like density of an extra trimer (Supplementary Fig. 4b). Only 164,259 particles from a 3D class that clearly exhibited the trimer doublet were selected for further processing. Particle images were re-extracted from micrographs into 256-pixel squared boxes without down-sampling. Because of the clear 2-fold

symmetry of the 3D class, C2 symmetry was imposed in subsequent steps. The initial 3D refinement resulted in a 3D reconstruction at 4.78 Å at 0.143 FSC. Then, CTF refinement for magnification anisotropy, optical aberrations, and per-particle defocus was performed on each optical group, followed by Bayesian particle polishing. These procedures were repeated three times. Next, the aligned particles were 3D-classified without further image alignment, from which 42,854 particles that belong in classes with detailed structural features were selected (Supplementary Fig. 4b). These particles were re-extracted into 512-pixel squared boxes to include the delocalized signal from highly defocused images (up to 5 μm underfocus). 3D refinement, CTF refinement, and particle polishing were repeated twice. The resolution of the final 3D reconstruction was 3.93 Å at 0.143 FSC (Supplementary Fig. 4d). 3D resolution anisotropy of the reconstruction was calculated using the 3DFSC server (https://3dfsc.salk.edu/)[39]. The resulting estimate indicates global resolution at 4.55 Å at 0.143 FSC and sphericity of 0.769 (Supplementary Fig. 4e).

For the D13 tubular assembly, 7,621 movies were motion-corrected followed by CTF estimation. Micrographs with poor estimated maximum resolution and severe astigmatism were discarded, based on the CTFFIND4 calculation. From 7,529 micrographs, start and end coordinates of the tubes were manually selected, and helical segments with a 167.5-Å interbox distance (5 × the estimated helical rise of 33.5 Å) were binned by 4 and extracted into 256-pixel squared boxes. A total of 194,960 particle images were subjected to 2D classification, from which 129,652 images belonging to class averages with good structural details were selected. Helical parameters were estimated by manual analysis of the helical layer-line pattern from the 2D class averages (Supplementary Fig. 6d). The images were classified in 3D using an artificial canonical helix reference consisting of spheres generated from helical parameters (77° twist, 33.5 Å rise with the command 'relion_helix_toolbox --simulate helix -o ref.mrc --subunit_diameter 140 --cyl_outer_diameter 900 --angpix 5.6 --rise 33.5 --twist 77 --boxdim 256'). The particles that belong to each class were re-extracted into 512-pixel squared boxes after binning by a factor of 2, followed by 3D refinement. Based on the detailed structural features of D13 trimers in the tubes and the nominal resolutions of the reconstructions, a total of 75,070 particle images from four 3D classes were selected and combined for the final refinement (Supplementary Fig. 6b). After 3D refinement, CTF refinement and Bayesian particle polishing were performed, followed by reconstruction with Ewald sphere correction[32,40]. The final map resolution was 7.3 Å at 0.143 FSC. Particles were re-extracted into 1024-pixel squared boxes and a 3D reconstruction was generated prior to symmetry expansion and signal subtraction. From the helical reconstruction, 5 asymmetric units were symmetry-expanded based on the final estimated helical operator (76.98° twist, 33.86 Å rise). For signal subtraction, a volume segment that corresponds to a ring of six trimers in the tube was generated from the reconstruction using UCSF Chimera. The map was then used to create a binary mask embracing six trimers, which was applied to the symmetry-expanded stack of particle images. Signalsubtracted sextet images were subjected to 2D class averaging using cisTEM[41], and 303,052 particles from good class averages were selected for autorefinement in 3 classes (Supplementary Fig. 6e, f). A total of 247,311 particles from 3D classes with detailed structural features were selected for final refinement in cisTEM. The resulting half-maps were examined and postprocessed using Relion 3.1. Final resolution of the map was 3.87 Å at 0.143 FSC (Supplementary Fig. 6g, h).

For Cryo-ET data, movie frames were aligned and summed using MotionCor2. Images were phase-flipped using ctfphaseflip[42] and the tilt series was aligned based on gold fiducial markers. Tomograms were reconstructed from images that were binned by 2 using the simultaneous iterative reconstruction technique (SIRT) implemented in IMOD[43].

**Protein structure modeling**. 3D reconstructions were subjected to density modification with the ResolveCryoEM tool[44] in the Phenix software suite prior to model refinement. All coordinate refinements were performed using the real-space refinement routine in Phenix[45]. The X-ray crystal structure of D13 (PDB ID 6BEI) was manually fitted into the cryo-EM map of our D13 trimer in UCSF Chimera and used as a reference model for atomic coordinate refinement in Phenix, while enforcing a noncrystallographic symmetry constraint. The resulting refined trimer model was then used as a reference to refine models of the trimer doublet and sextet. These refined models were manually inspected and adjusted in Coot[46], followed by final refinement with Phenix. The quality of the final models and map-to-model correlations were calculated using Phenix' Cryo-EM validation and Mtriage tools[47].

**Reporting summary**. Further information on research design is available in the Nature Research Reporting Summary linked to this article.

## Data availability

Atomic coordinates of cryo-EM based models have been deposited under PDB IDs 7VFD (wtD13 trimer), 7VFE (His₆-tagged D13 trimer), 7VFF (D13₁₈₋₅₄₇ trimer), 7VFG (D13 trimer doublet) and 7VFH (D13 trimer sextet). Cryo-EM maps have been deposited in the EM database under accession codes EMD-31949 (wtD13 trimer), EMD-31950 (His₆-tagged D13 trimer), EMD-31951 (D13₁₈₋₅₄₇ trimer), EMD-31952 (D13 trimer doublet),

EMD-31953 (D13 trimer tubular assembly) and EMD-31954 (D13 trimer sextet). Raw image data are available from the authors upon reasonable request.

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

## Acknowledgements

We thank Dr. Fasséli Coulibaly for critical reading of our manuscript. We are grateful to Dr. Steven Aird for technical editing. We appreciate the support from the Scientific Computing and Data Analysis Section (OIST RSD). We acknowledge the Scientific Imaging Section (OIST IMG) for use of the cryo-EM facility. This work was supported by the Platform Project for Supporting Drug Discovery and Life Science Research (BINDS) from AMED under grant number JP18am0101076 (to M.W.), and by JSPS KAKENHI grants JP20K06581 (to H.M.) and 21K06039 (to J.H.). M.W. was supported by direct funding from Okinawa Institute of Science and Technology Graduate University.

## Author contributions

J.H. conceived and designed experiments. J.H. and T.K. performed molecular cloning and protein purification. J.H. created mutants. J.H. and M.W. collected cryo-EM data. J.H. processed cryo-EM data. H.M. and J.H. carried out atomic modeling and refinement into cryo-EM maps. M.W. supervised the project and provided advice on cryo-EM and image processing. J.H. and M.W. wrote the manuscript. All authors analyzed results and contributed to writing the paper.

## Competing interests

The authors declare no competing interests.
