## [Peer Review File · Nature Communications]

Assembly mechanism of the pleomorphic immature poxvirus scaffoldREVIEWER COMMENTS

Reviewer #1 (Remarks to the Author):

In this manuscript the authors analyse the vaccinia virus (VACV) scaffold protein, gene product of D13, expressed in vitro by cryoEM and image processing. When expressed in vitro the protein is known to form trimers with a central tunnel structure that accommodates the extreme N-terminus of its binding partner A17. In infected cells the trimers arrange in arrays of hexamers forming a so-called 'honeycomb' like lattice on the convex side of the viral membrane. Similar to the formation of clathrin cages does the D13 protein form a single layer of the lattice on the surface of the membrane, making it bend and assuring the formation of the viral spherical membrane with a constant diameter of roughly 350nm. The present manuscript asks how trimers interact to form hexamers expressing purified protein and mutant protein. This question is both interesting and important and should be addressed 'initially' in vitro.

From a technical point of the view, the image processing, the data are sound. The maps allow a true evaluation of the data as does the description of the processing pipelines.

A few comments, however, remain that should be addressed:

1. In the samples without the graphene oxide films the proteins sticks to the air-water interface forming the honeycomb like lattices - this is a problem for the trimer picking/processing as it causes a preferred orientation issue. Why the tilted acquisition was performed on grids without the graphene oxide film if Sup. fig. 2 shows how the graphene can help - it seems an odd choice not to use it for all data acquisition. While tilting helps with the preferred orientation the problem of too clustered trimers remains, not to mention that at the tilted image the clustered particles will partially overlap in the acquired images which will complicate the processing (and if too many of those are used it can even introduce some artifacts into the reconstruction).
2. In Supp. 2e the FSC curve does not cross the zero. This might be caused by too tight mask used for FSC computation - a proper mask should be used and the corresponding FSC resolution reported.
3. In Supp. Fig. 4. How did the authors identified the "good" classes? The visual inspection of the presented classes does not justify the authors' choice as the classes do not seem that similar and the diameters also differ.

Major scientific comments:

1. A fundamental question is what we learn about the biology of D13, how it forms the typical lattice observed in infected cells that deform membranes to form a sphere with a rather constant diameter. The cryoEM data dissect two interfaces in D13, in its head domain and at the base, that mediate inter-trimer interaction, resulting in trimer doublets. However, the full-length protein does not make a honey-

comb lattice under the in vitro conditions used. Honey comb like lattices are formed only when the N-terminus is 'inactivated' either by tagging it with a His-tag or if it is deleted altogether. The truncated D13 then forms helical tubes with a honey comb arrangement consisting of several superimposed D13 layers. I am not sure if these structures, in particular the helical tubes, have any functional relevance. I would have much preferred to see D13-structures (artificially) bound to membranes in vitro in particular since the first author has previously shown that this is feasible (Hyun et al., 2011; doi:10.1371/journal.ppat.1002239). Why did the authors not attempt to do this, or why were the putative interactions of the his-tagged construct, forming spherical structures not analysed? Were the structures obtained to heterogenous? These points should be discussed.

2. In situ the situation might be more complex; D13 typically forms a single scaffold layer on the membrane that bends it with a constant diameter, rather than tubes. Multiple layers of D13 are only made under mutant conditions such as the D513G mutation (Szajner et al., 2005; doi/10.1083/jcb.200504026). The authors emphasize (on more than one occasion) that D13 self assembles without need of accessory proteins and that honey comb formation in infected cells requires membrane binding (I guess this assumption is based on a model presented in the publication by Garriga et al., 2018). Data acquired in infected cells (Szajner et al., 2005; doi/10.1083/jcb.200504026 and Chlanda et al., 2009; DOI 10.1016/j.chom.2009.05.021) argue against this. Szajner et al. provided data, that in infected cells scaffold formation requires the synthesis of viral late proteins. Chlanda et al. showed, based on 3D-EM data of infected cells, that D13 can form small honey comb patches that are not associated with membranes. Hence, it is mandatory to modify the introduction and the discussion accordingly incorporating and discussing these results.

Reviewer #2 (Remarks to the Author):

The manuscript by Hyun et al takes a closer look (2.3A) at the poxvirus scaffolding protein D13 using cryo-EM of purified protein. The report is well written and provides further insight into the form and function of this essential and interesting virus protein. This reviewer has only 1 minor issue to fix. Centrifugations should be reported as RCF or x g (Line 311). RMPs are not helpful in determining how much force is required to pellet the trimers.

Reviewer #3 (Remarks to the Author):

This manuscript by Hyun et al presents a comprehensive study of a remarkable assembly mechanism evolved by large DNA viruses with internal membranes. By contrast with well described enveloped viruses, these viruses do not gain their internal membrane by budding through cellular organelles but assemble it in situ. This study advances our molecular understanding of the scaffold involved in remodelling of the growing viral membrane during morphogenesis. Comparison of several structures proposed to mimic assembly intermediates suggest a molecular mechanism for curvature formation and a regulatory role for the N-terminal helix, which had not been identified in previous structural studies or in-vitro assembly experiments.

This model is a striking example of an alternative morphogenesis path compared to typical icosahedral capsids or enveloped viruses produced by budding. Using cryo-EM and in vitro assembly, the study achieves the challenging task of characterising an assembly process that relies on significant plasticity in inter-capsid interactions. The multiple interaction surfaces identified in this study explain the formation of a continuous curvature that departs from “angular” icosahedral structures that have been elucidated for other large DNA viruses.

The findings are novel and relevant to the assembly of lipid-containing large/giant viruses. They are well supported by the data presented and I only have few comments listed below.

Main comments

1. Inter-trimer interactions (p.5).

To strengthen the biological relevance of these interactions, it would be useful to (i) compare them to the low resolution images/modelling of native IVs (Heuser et al, 2005; may be Szajner, 2005), and (ii) discuss the sequence conservation of the residues and interfaces involved.

2. “...assembled exclusively into spherical particles that mimic the size and shape of the authentic VACV IV scaffold” (p. 6, l.145-148).

This should be rephrased or shown quantitatively by (i) providing representative low magnification fields of view showing more IV-like objects and the absence of other assembled forms of D13 in these conditions; and (ii) providing a comparative analysis of the sizes and organisation of these objects vs. IVs as described by Heuser, 2005. Related to this point, tomography shows the presence of a double layer or concentric shells that does not look like typical IVs. It would be worth having the authors' interpretation of these objects in the movie description.

3. N-terminal helix role.

- “... additional charges on the N-terminal His-tag that perturb the hydrophobic environment that stabilizes this interaction in native, untagged protein” (p.6, l. 152-156).

The His-tag is located much further out and there are no obvious clashes. It is more likely that the linker (e.g. through Y-4 and F-3) - not the poly-histidine sequence - displaces the helix directly making hydrophobic contact as seen in some of the crystal structures where one of the 3 sites is occupied by the linker.

- "... thereby confirming the essential function of the N-terminal tail helix for D13 self-assembly." (p. 6, l161-162).

This is confusing since self-assembly happens when the N-ter helix is displaced or deleted. So it is clearly not essential to self-assembly per se, but, as shown for the first time here, it inhibits assembly and possibly regulates the nature of the assembly (IV-like vs. tubes vs. honeycomb lattice) when displaced. Would the presence of the N-ter helix create clashes incompatible with the inter-trimer interfaces described here (doublets, helical)? It is not directly shown or discussed (Fig. 2c suggests there is sufficient space).

4. A couple of questions relate to Figure 5 and the discussion:

- Is there evidence for the building blocks shown in 5b? All the structures described earlier in the manuscript appear to have a different arrangement and, indeed, mutation of residues in the jelly-roll base is described as inhibiting assembly.

- What exactly is represented in 5d? It would make sense that this modelling incorporates data from the cryo-ET reconstruction and the various inter-trimer assemblies described earlier but it is not clear if and how this has been done here. This analysis would benefit from more details about the actual diameters / curvature of native IVs, IV-like objects produced here and the models using mode 1/2 and pentameric assembly of D13.

Minor comments:

p4, l. 82: "exceeds the resolution of even the best available crystal structures."

Although comparing cryo-EM and crystal structure resolutions remains contentious, the nominal resolutions support this conclusion. A couple of comments: (i) it would be interesting to know if water molecules are visible in the reconstruction and, if so, how many compared to the crystal structures; (2) if referring to the "best crystal structures", the 2.55Å one should be cited here (6BEB).

p.4, l. 94: "In our structure...". This suggest the contact surface differs in the new structure compared to the crystal structures previously determined. It would be very interesting if that's really the case and differences should be described in more details.

Supplementary Figure 1: "concentric rings". I only see rings of trimers. I do not see what "concentric" refers to in this context.

Supplementary Fig. 6 (left panels) is not clear. It seems to show honeycomb lattices but is described as low salt (i.e. should lead to IV formation for His6-D13 or tubes for D13_18-547). Are these flattened IVs due to the negative staining process?

In the discussion (p.9, l. 254), the local symmetries involving Modes I and II (and probably other modes) cannot be said to follow the quasi-equivalence principle as defined by Caspar and Klug.

RESPONSE TO THE REVIEWERS

Reviewer #1 (Remarks to the Author):

In this manuscript the authors analyse the vaccinia virus (VACV) scaffold protein, gene product of D13, expressed in vitro by cryoEM and image processing. When expressed in vitro the protein is known to form trimers with a central tunnel structure that accommodates the extreme N-terminus of its binding partner A17. In infected cells the trimers arrange in arrays of hexamers forming a so-called 'honey-comb' like lattice on the convex side of the viral membrane. Similar to the formation of clathrin cages does the D13 protein form a single layer of the lattice on the surface of the membrane, making it bend and assuring the formation of the viral spherical membrane with a constant diameter of roughly 350nm. The present manuscript asks how trimers interact to form hexamers expressing purified protein and mutant protein. This question is both interesting and important and should be addressed 'initially' in vitro.

From a technical point of the view, the image processing, the data are sound. The maps allow a true evaluation of the data as does the description of the processing pipelines.

A few comments, however, remain that should be addressed:

1. In the samples without the graphene oxide films the proteins sticks to the air-water interface forming the honeycomb like lattices - this is a problem for the trimer picking/processing as it causes a preferred orientation issue. Why the tilted acquisition was performed on grids without the graphene oxide film if Sup. fig. 2 shows how the graphene can help - it seems an odd choice not to use it for all data acquisition. While tilting helps with the preferred orientation the problem of too clustered trimers remains, not to mention that at the tilted image the clustered particles will partially overlap in the acquired images which will complicate the processing (and if too many of those are used it can even introduce some artifacts into the reconstruction).

Thank you for this comment. We were aware of the challenges presented by tilted data acquisition. However, doublet particles on graphene oxide (GO) were of limited angular orientations, providing either top or side views mostly. Without additional data of tilted specimen, which provided additional angular orientations of the particles, the 3D reconstruction attempts were never successful. Doublet particles without GO film adopted a wider angular range of views and enabled successful 3D reconstruction.

This description has been added to the figure legend of Sup. Fig. 4.

2. In Supp. 2e the FSC curve does not cross the zero. This might be caused by too tight mask used for FSC computation - a proper mask should be used and the corresponding FSC resolution reported.

Thank you for pointing this out. The particle images used in the final reconstruction were down-sampled by a factor of 2, hence truncated at half Nyquist (2.07 Å) and caused the curve to be cut off before reaching zero. We plotted the FSC using the reconstruction produced without down-sampling that represents the full resolution range. The FSC curve is now observed reaching zero, as expected (below).

However, please understand that Sup. Fig 2e is kept unchanged because the cryo-EM map (down-sampled) had already been deposited to the EMDB and we would have to resubmit the map to match the FSC curve if it were updated in the manuscript. The legend of SI Fig. 2e has been updated accordingly.

3. In Supp. Fig. 4. How did the authors identified the "good" classes? The visual inspection of the presented classes does not justify the authors' choice as the classes do not seem that similar and the diameters also differ.

Thank you for the comment. We refined particles that belong in each 3D class and chose the classes with detailed structural features of the trimer, based on (1) visual inspection and (2) nominal resolution. This part of the data processing had been omitted in the submitted figure due to space limitation. We now replaced Supplementary Figure 6b with independently refined classes and their nominal resolution to clarify this.

Yes, the diameter and helical operators differ between some of the classes and our attempt to further segregate or combine classes did not improve the resolution of the map. The combination of the images from the selected classes, which have nearly identical helical parameters, led to the best map resolution.

We have now replaced the ambiguous term “good classes” with a more specific description as mentioned above (lines 502-506)

Major scientific comments:

1. A fundamental question is what we learn about the biology of D13, how it forms the typical lattice observed in infected cells that deform membranes to form a sphere with a rather constant diameter. The cryoEM data dissect two interfaces in D13, in its head domain and at the base, that mediate inter-trimer interaction, resulting in trimer doublets. However, the full-length protein does not make a honey-comb lattice under the *in vitro* conditions used. Honey comb like lattices are formed only when the N-terminus is 'inactivated' either by tagging it with a His-tag or if it is deleted altogether. The truncated D13 then forms helical tubes with a honey comb arrangement consisting of several superimposed D13 layers. I am not sure if these structures, in particular the helical tubes, have any functional relevance. I would have much preferred to see D13-structures (artificially) bound to membranes *in vitro* in particular since the first author has previously shown that this is feasible (Hyun *et al.*, 2011; doi:10.1371/journal.ppat.1002239). Why did the authors not attempt to do this, or why were the putative interactions of the his-tagged construct, forming spherical structures not analysed? Were the structures obtained to heterogenous? These points should be discussed.

Thank you for this insightful comment. Previous results indeed demonstrated membrane remodeling by D13 assembly, showing continuous honeycomb-like lattice on lipid membrane (Hyun *et al.* 2011). However, the experiment utilized specific interactions between N-terminal His₆-tag and Ni²⁺ on a functionalized lipid surface. We therefore considered that the proteoliposome approach may introduce artifacts where the intrinsic curvature of the lipid vesicles may hamper clear interpretation of the curvature generated by D13 assembly alone.

As shown in our manuscript, D13 with 'inactivated' N-terminus assembles into continuously curved spheres that closely resembles the immature VACV virion scaffold without the presence of additional lipid. We emphasize this results in the revised manuscript by replacing the image of the spherical particles in the Fig. 3b and 3c (right column) with much clearer cryo-EM images, and by adding a representative cryo-electron tomographic reconstruction of the spherical particles (Sup. Fig. 5). Investigating the interaction between D13 trimers in the context of the spherical assembly in more detail was attempted by cryo-ET and subtomogram averaging, but we could not reach the level of resolution of our single particle reconstructions required for unambiguous insights.

The tubular assembly based on the N-terminal truncation mutant may not directly be biologically relevant. However, we observed coexistence of the tubular assembly products and spherical particles under our *in vitro* assembly conditions, and we believe that these assembly products are variants of curved assembly. Therefore, tubular assembly must represent part of the curvature-forming property of D13 assembly. This point is now discussed in greater detail in the main text (lines 174-176 and legend of Fig. 3).

2. In situ the situation might be more complex; D13 typically forms a single scaffold layer on the membrane that bends it with a constant diameter, rather than tubes. Multiple layers of D13 are only made under mutant conditions such as the D513G mutation (Szajner et al., 2005; doi/10.1083/jcb.200504026). The authors emphasize (on more than one occasion) that D13 self-assembles without need of accessory proteins and that honey comb formation in infected cells requires membrane binding (I guess this assumption is based on a model presented in the publication by Garriga et al., 2018). Data acquired in infected cells (Szajner et al., 2005; doi/10.1083/jcb.200504026 and Chlanda et al., 2009; DOI 10.1016/j.chom.2009.05.021) argue against this. Szajner et al. provided data, that in infected cells scaffold formation requires the synthesis of viral late proteins. Chlanda et al. showed, based on 3D-EM data of infected cells, that D13 can form small honey comb patches that are not associated with membranes. Hence, it is mandatory to modify the introduction and the discussion accordingly incorporating and discussing these results.

Thank you again for insightful comments. We are aware of previous works by Szajner *et al.* (2005), Heuser (2005), Chlanda *et al.* (2009) and more recent work by Weisberg *et al.* (2017), who are all showing the complex nature and concerted involvements of viral late proteins in immature virion formation. We fully appreciate these studies which have provided a strong foundation for our work.

As you have pointed out, the description of the spherical D13 assembly in the Introduction/Discussion needs to address the complexity of the *in situ* situation. We have accordingly modified the manuscript (lines 43-52 and 223-227). We are now also providing a modified Figure 3b showing spherical D13 assembly products by cryo-EM and a cryo-tomography (CET) reconstruction (Sup. Fig. 5). These data strongly support that D13 self-assembly mimics the authentic scaffold and plays a major role in determining the morphology of the IV.

Reviewer #2 (Remarks to the Author):

The manuscript by Hyun et al takes a closer look (2.3A) at the poxvirus scaffolding protein D13 using cryo-EM of purified protein. The report is well written and provides further insight into the form and function of this essential and interesting virus protein. This reviewer has only 1 minor issue to fix. Centrifugations should be reported as RCF or x g (Line 327). RMPs are not helpful in determining how much force is required to pellet the trimers.

Thank you very much for your comment. The units have been updated accordingly (line 332).

Reviewer #3 (Remarks to the Author):

This manuscript by Hyun et al presents a comprehensive study of a remarkable assembly mechanism evolved by large DNA viruses with internal membranes. By contrast with well described enveloped viruses, these viruses do not gain their internal membrane by budding through cellular organelles but assemble it in situ. This study advances our molecular understanding of the scaffold involved in remodelling of the growing viral membrane during morphogenesis. Comparison of several structures proposed to mimic assembly intermediates suggest a molecular mechanism for curvature formation and a regulatory role for the N-terminal helix, which had not been identified in previous structural studies or in-vitro assembly experiments.

This model is a striking example of an alternative morphogenesis path compared to typical icosahedral capsids or enveloped viruses produced by budding. Using cryo-EM and in vitro assembly, the study achieves the challenging task of characterising an assembly process that relies on significant plasticity in inter-capsid interactions. The multiple interaction surfaces identified in this study explain the formation of a continuous curvature that departs from “angular” icosahedral structures that have been elucidated for other large DNA viruses.

The findings are novel and relevant to the assembly of lipid-containing large/giant viruses. They are well supported by the data presented and I only have few comments listed below.

Thank you very much for your review.

Main comments

1. Inter-trimer interactions (p.5).

To strengthen the biological relevance of these interactions, it would be useful to (i) compare them to the low resolution images/modelling of native IVs (Heuser et al, 2005; may be Szajner, 2005), and (ii) discuss the sequence conservation of the residues and interfaces involved.

Thank you for helpful suggestions.

(i) We revisited Heuser (2005) and Szajner *et al.* (2005) and made sure that our *in vitro* assembly products closely resemble the authentic VACV IV scaffold as shown by inclusion of the deep-etch EM figure (SI Fig. 5). We have added a figure from Heuser (2005) in Supplementary Data Fig. 5b for direct comparison with our cryo-electron tomography reconstruction of the spherical assembly product.

(ii) We performed sequence alignment with D13 homologues from species of a number of poxvirus genera, including both chordopoxvirus and entomopoxvirus subfamilies. We have now added an additional figure (Sup. Fig. 9) about sequence conservation of residues involved in scaffold assembly and provide discussion in the main text (lines 249-252).

2. “...assembled exclusively into spherical particles that mimic the size and shape of the authentic VACV IV scaffold” (p. 6, l.145-148).

This should be rephrased or shown quantitatively by (i) providing representative low magnification fields of view showing more IV-like objects and the absence of other assembled forms of D13 in these conditions; and (ii) providing a comparative analysis of the sizes and organisation of these objects vs. IVs as described by Heuser, 2005. Related to this point, tomography shows the presence of a double layer or concentric shells that does not look like typical IVs. It would be worth having the authors' interpretation of these objects in the movie description.

Thank you again for helpful suggestions.

(i) We have replaced Fig. 3b showing additional IV scaffold-like assembly products. The description in the main text has been modified accordingly (lines 155-162). Presence of a few multi-layered objects and incomplete particles are also described in the figure legend of Fig 3 (lines 580-581).

(ii) We have now added Sup. Fig. 5, showing a cryo-electron tomography reconstruction in stereoscopic iso-surface representation, as well as z-slices of the reconstruction to describe the morphology of spherical particles. An image from Heuser (2005) is added to the figure for direct comparison.

The presence of multiple-layered shells had been described for the D13_{D513G} mutation and for a temperature-dependent lethal mutant that induces accumulation of multi-layered scaffolds (Szajner *et al.* 2005). In our *in vitro* assembly, such extensive multilayers are not observed (usually double layer). We speculate that opposite charges on head and base of D13 as shown in Supplementary Fig. 7 may drive the stacking of the shells. This description has been added to the Supplementary Movie description.

3. N-terminal helix role.

- “... additional charges on the N-terminal His-tag that perturb the hydrophobic environment that stabilizes this interaction in native, untagged protein” (p.6, l. 152-156).

The His-tag is located much further out and there are no obvious clashes. It is more likely that the linker (e.g. through Y-4 and F-3) - not the poly-histidine sequence - displaces the helix directly making hydrophobic contact as seen in some of the crystal structures where one of the 3 sites is occupied by the linker.

We appreciate your comment. We have modified the description in the main sentence by referring to the N-terminal linker seen in previous crystal structure (lines 164-168).

- “... thereby confirming the essential function of the N-terminal tail helix for D13 self-

assembly.”(p. 6, l161-162).

This is confusing since self-assembly happens when the N-ter helix is displaced or deleted. So it is clearly not essential to self-assembly per se, but, as shown for the first time here, it inhibits assembly and possibly regulates the nature of the assembly (IV-like vs. tubes vs. honeycomb lattice) when displaced. Would the presence of the N-ter helix create clashes incompatible with the inter-trimer interfaces described here (doublets, helical)? It is not directly shown or discussed (Fig. 2c suggests there is sufficient space).

We agree that the sentence is misleading description of the role of N-terminal helix and have removed it from the main text.

As you have pointed out, the space at the base-to-base interface appears large enough to occupy 2 copies of N-terminal helix, but there is a clash between sidechains (new additional Sup. Fig 3b). This is now mentioned in the legend of Supplementary Fig. 3b.

In our study, we provide a general mechanism, how displacement of the N-terminal helix induces scaffold formation. However, the detailed regulatory mechanism by the N-terminal helix in its native biological environment may involve other viral proteins or interaction with viral membranes and would require further investigation.

4. A couple of questions relate to Figure 5 and the discussion:

- Is there evidence for the building blocks shown in 5b? All the structures described earlier in the manuscript appear to have a different arrangement and, indeed, mutation of residues in the jelly-roll base is described as inhibiting assembly.

We agree that the partial doublet in Fig. 5b was misleading. Our initial idea was to represent a partially assembled D13 that harbors only head-to-head interaction when the base interaction is blocked by intact N-terminal helix. We have now removed 5b and reorganized Fig. 5 to simplify the message.

- What exactly is represented in 5d? It would make sense that this modelling incorporates data from the cryo-ET reconstruction and the various inter-trimer assemblies described earlier but it is not clear if and how this has been done here. This analysis would benefit from more details about the actual diameters / curvature of native IVs, IV-like objects produced here and the models using mode 1/2 and pentameric assembly of D13.

We appreciate this valid criticism. The expanded honeycomb lattice in Fig 5d was generated by overlapping the D13 trimer sextet model. As described in the figure legend, the attempt failed to build a plausible curvature model and the purpose of the figure was only meant for visualization of our hypothesis of VACV scaffolding.

We have now replaced the figure with a new cryo-electron tomography reconstruction that demonstrates size and morphology of the IV-like particle. However, we did not model hexameric/pentameric assembly into the tomography data, since the low resolution could only visualize the honeycomb lattice. A much clearer depiction would require further study using subtomogram averaging, which is beyond the scope of the present paper.

Minor comments:

p4, l. 82: “exceeds the resolution of even the best available crystal structures.”

Although comparing cryo-EM and crystal structure resolutions remains contentious, the nominal resolutions support this conclusion. A couple of comments: (i) it would be interesting to know if water molecules are visible in the reconstruction and, if so, how many compared to the crystal structures; (2) if referring to the “best crystal structures”, the 2.55Å one should be cited here (6BEB).

Thank you for the comment. We have removed this claim and simply state the map resolution (line 90)

In our model, we counted 352 water molecules (compared to 551 waters in 6BEB).

p.4, l. 94: “In our structure...”. This suggest the contact surface differs in the new structure compared to the crystal structures previously determined. It would be very interesting if that's really the case and differences should be described in more details.

The sentence aimed at describing the structure of the N-terminal helix with its hydrophobic pocket, which has not been extensively discussed in previous publications, but not to emphasize the uniqueness of our cryo-EM structure.

To address your point, we have now compared our cryo-EM structure with previously determined crystal structures. The difference was marginal. The description has now been added to the main text (lines 104-107), and additional Sup. Fig. 3a is provided.

Supplementary Figure 1: “concentric rings”. I only see rings of trimers. I do not see what “concentric” refers to in this context.

Thank you for pointing it out. The word “concentric” was removed.

Supplementary Fig. 6 (left panels) is not clear. It seems to show honeycomb lattices but is described as low salt (i.e. should lead to IV formation for His6-D13 or tubes for D13_18-547). Are these flattened IVs due to the negative staining process?

Thank you for the comment. The assembly products are typically flattened by negative staining and dehydration. We used images of negatively stained particles for consistency with the rest of the figures. For clarity, we have now replaced the images of assembly products with cryo-electron micrographs (Fig. 3b,c and Sup. Fig. 8).

In the discussion (p.9, l. 254), the local symmetries involving Modes I and II (and probably other modes) cannot be said to follow the quasi-equivalence principle as defined by Caspar and Klug.

Thank you for pointing this out – we have now removed this statement (line 272).

REVIEWERS' COMMENTS

Reviewer #1 (Remarks to the Author):

This reviewer found the revision and rebuttal convincing. The data can now be published.

Reviewer #3 (Remarks to the Author):

The authors have modified the manuscript to answer all previous queries and comments satisfactorily. I commend them on a fine study.